# ASSOCIATION RULES IN QUBO SAMPLES AND WHERE TO FIND THEM

## ABSTRACT

There are sometimes strong associations between variables in the samples to a Quadratic Unconstrained Binary Optimization (QUBO) problem. A natural question arises to us: Are there any value in these association? We study max-cut problem and observe that association can be represented as rules to simplify QUBO problem. Classical and quantum annealers work better when the problem size is smaller. To effectively and efficiently find associations between variables, we adapt traditional association rule mining in the case of QUBO samples and propose a Fast Association Rule Mining algorithm (FARM) specifically for mining QUBO samples. We also propose strategies and a workflow to select and apply promising rules and simplify QUBO problems. We evaluate our method on D-Wave Quantum Annealer as well as Fujitsu Digital Annealer. The experiments demonstrate the utility of FARM as a visualisation tool for understanding associations in QUBO samples. The results also demonstrate the potential of our method in closing the gap between samples and ground truth. The source code will be disclosed to the public if the manuscript is accepted.

## 1 INTRODUCTION

Many combinatorial optimization problems can be formulated as a quadratic unconstrained binary optimization problem (QUBO) Lucas (2014), Glover et al. (2018). QUBO corresponds naturally to the transverse Ising model and benefits from the speed up by quantum annealing Kadowaki & Nishimori (1998). The combination of QUBO and Quantum annealing Kadowaki & Nishimori (1998) has proved its utility in various applications, such as Traveling Salesman Problem (TSP) Martoňák et al. (2004), Graph colouring Titiloye & Crispin (2011) Portfolio Optimization Venturelli & Kondratyev (2019) and Resource Scheduling Problem Ikeda et al. (2019).

Annealing, e.g., simulated annealingKirkpatrick et al. (1983), is a family of probabilistic methods for optimising the variables of a system (e.g. minimising a function). In annealing, we heat the system to a high-temperature level and gradually bring down the temperature. Variables in the system gradually lose their energy and eventually sit in a low-energy state. One round of heating and cooling, i.e., an annealing process, produces one sample, which is a configuration of all variables. The annealing process is random. The energy distribution of the samples follows a Boltzmann distributionNelson et al. (2022), i.e., the system is more likely to be in a lower energy state. Quantum annealing works similarly, except that it makes use of quantum mechanics.

We can use annealing for optimisation in a sampling manner. While we cannot find optimal solution in one shot, one possible way to obtain a promising solution is to collect more samples. However, the marginal profit of "more samples" decreases as we collect more samples. Besides, access to quantum devices is expensive. We are looking for a more efficient method to make additional samples more productive. The main idea of the paper is that we examine existing samples to discover interesting rules and simplify the original QUBO problem. More specifically, if most of the variables agrees on certain decision, then most likely these variables have been solved correctly. We should then solve the remaining variables that we are less certain of.

There are some earlier works in this domain, with their own limitations. Lewis & Glover (2017) invented a few rules to transform the underlying QUBO structure into an equivalent graph with fewer nodes and edges. However, such rules are hand-crafted. A question of interest is whether we can discover such rules that could be specific to a particular application. Chardaire et al. (1995); Karimi

& Rosenberg (2017); Irie et al. (2021) utilise sample persistency, which concerns the persistency of individual variables, and omits the associations between variables. These methods do not work well if persistency is not explicitly presented in samples. For example, in a max-cut problem, each sample and its flipped version are equivalent, both having equal chances of presence in sampling. Therefore, no variable in max-cut is going to be persistent. Other problems like number partitioning, graph colouring, and TSP, also share a similar feature.

In this paper, we propose to use heuristic method to discover association rules between variables. The concept of traditional Assocation Rule Mining (ARM) fits very well with our purpose, as it is originally designed for discovering associations between items in Market Basket Analysis (MBA) Agrawal et al. (1993). Through adapting traditional ARM in the mining task of QUBO samples, we can discover associations between variables, reveal hidden persistency in samples, and simplify QUBO problems accordingly. By solving a smaller QUBO problem, we stand a better chance of obtaining more promising results. Our contributions can be summarised as follows.

- We adapt association rule mining to the case of mining rules from QUBO samples. We introduce the concept of equivalent alternatives in the definition to couple with the special requirements of applications like max-cut.
- We explore effective ways to select and apply rules to reduce QUBO problems.
- We propose Fast Association Rule Mining (FARM) algorithm for mining QUBO samples, to efficiently discover rules of wide range of support (typically 10-90%).

In the experiments, we evaluate our definitions and methods on two solvers, D-Wave Pegasus Quantum Annealer Willsch et al. (2022) (QA) and Fujitsu 2nd Gen Digital Annealer Aramon et al. (2019) (DA). We use max-cut problems as a case study and include two datasets with different topologies in the experiment. Results on QA suggest that FARM can serve as a visualisation tool to help understand associations between variables. By fixing variables according to the discovered rules, we can simplify a QUBO problem and find more promising solutions in a second run of solvers. Results on DA suggest that classical annealing-based solvers produce very low level of persistency on large problems, and receives very limited benefits from the proposed method. Comparing the results on QA and DA, we know that association rule has more potential in QA.

The rest of the paper is organised as follows. Section 2 reviews existing literature and describes the position of this work. Section 3 introduces the concept of association rule in the case of mining QUBO samples and presents a workflow and a few strategies that apply the discovered rule to simplify a QUBO problem. Section 4 describes the assumption and design ideas of Fast Association Rule Mining (FARM) for QUBO samples. Section 5 quantitatively compares our methods with a representative baseline method. Section 6 draws the conclusion.

## 2 RELATED WORK

### 2.1 OPTIMISATION

A QUBO minimisation problem can be written in the form of $\min_{x \in \{0,1\}^n} x^T Q x$, where $x$ is a vector of decision variables, $Q$ is the QUBO matrix. Sometimes, the answers to some variables are trivial, such that their value remains unchanged in the optimal solution, regardless of the values taken by other variables. This is called variable persistence in Boros et al. (2006). We can assign the values so that the variables are fixed. By this partial assignment, we effectively reduce the number of variables, and in turn, reduce or simplify the QUBO problem.

There are a few ways to identify such partial assignments. One way is to check the QUBO matrix against a set of rules Lewis & Glover (2016; 2017); Glover et al. (2018). These include manually crafted rules for assigning values to individual variables and a pair of variables. This method is designed as preprocessor to reduce the size of a problem before passing it to any solvers. The preprocessor "QPro" scans through a QUBO matrix and apply the rules in an iterative way until no further variables can be assigned values. Experiments suggest that QPro works effectively on a synthetic dataset with sparse connectivity and non-uniform distributions Glover et al. (2018) and scientific dataset Şeker et al. (2020), but failed to yield any reduction on the dense and uniformly distributed ORLIB 2500 variable problems Beasley (2010) and a real-world problem Kim et al.

(2020). We are aware of the limitation of manually crafted rules on practical problems. In this work, we are developing a heuristic method to automatically discover rules.

Such a partial assignment can also be identified by exploiting the dynamics in simulation-based optimisation heuristics. For example, Chardaire et al. (1995) focus on the simulated annealing algorithm and use the term "thermostatistical persistence" to describe the value of a variable that remains constant as the annealing temperature decreases. The author "fix" such variables so that the rest of the annealing process is guided and accelerated. Irie et al. (2021) use molecular dynamics simulation instead. The authors use the term "frozen" and "ambivalent" to refer to variables that obtain "persistence" at an early or late stage of molecular dynamics simulation. The frozen variables are the "precondition" of the original problem. The ambivalent variables remain in the reduced problem, the size of which is determined by a manually-chosen hyper-parameter. The utility of these methods is based on and restricted by the characteristics of specific optimisation heuristics. In our work, we develop a solver-independent method that can directly work on the samples or solutions of original problems.

It is also possible to identify partial assignments by directly studying samples or solutions of solvers. Karimi & Rosenberg (2017); Karimi et al. (2017) use the term "sample persistency" to describe a variable that has the same value in most samples or solutions. The authors conjecture that variables with persistent values are likely to maintain their values in the ground state of the system. Such variables can be included in partial assignment. Sample persistency has found its utility in jet clustering Delgado & Thaler (2022) and subsurface hydrological inverse analysis Golden & O'Malley (2021).

Sample persistency Karimi & Rosenberg (2017) is similar to the heuristic-based methods Chardaire et al. (1995); Irie et al. (2021), in the sense that they only concern the persistency of individual variables, and omit associations between variables. These methods will lose their utility if sample persistency is not explicitly presented in samples. Take max-cut problem as an example, each sample and its flipped version are equivalent to each other, such that you can always expect a persistency of 50% for all variables in a fair sampling. Other problems like number partitioning, graph colouring problem and Traveling Salesman Problem (TSP) also have such characteristics.

Association Rule Mining (ARM) is a good fit in solving the above mentioned issues. We are going to adapt ARM to find assocation between variables, which can be used to reveal the implicit persistency in the mining task for QUBO samples. However, ARM is originally not designed for mining QUBO samples.

## 2.2 ASSOCIATION RULE MINING

Association rule mining Agrawal et al. (1993) (ARM) finds strong correlations and frequent patterns from sets of items in the transaction databases. The original concept is for Market Basket Analysis (MBA), e.g., optimising the arangement of items on racks in a supermarket, such that the number of items purchased by customers are maximised. There have been various algorithms for ARM: the vanilla ARM algorithm AIS Agrawal et al. (1993), the pruning-enabled milestone, i.e., Apriori Agrawal et al. (1994), tree structured approaches, like FP-Growth Han et al. (2000) and RARM Das et al. (2001), approximated algorithms like ARMA Nayak & Cook (2001) and PARMARiondato et al. (2012). These algorithms assume that the transaction database is getting large (e.g. more than millions), whereas the size of each transaction is limited (e.g. an individual customer usually buys a few dozens of items in a single shopping trip). Take the FP-Growth algorithm as an example; it maintains a tree structure, the size of which scales very well with the number of transactions, but exponentially increases along with the average number of items in transactions. With high support rate and pruning technique, it eliminates a lot of computation and ensures effective mining.

In the mining task for QUBO samples we assume a limited number of samples (corresponds to transactions, e.g. a few hundreds), ever increasing number of variables (corresponds to items, e.g., a few thousands or even more) and a wide range of support. Even if we avoid tree construction by using the Apriori algorithm, the generation of candidates is also very expensive if we use a low support rate. Our assumptions are quite different from the traditional ARM applications and cause existing ARM algorithms to be computationally intractable for our mining task.

# 3 ASSOCIATION RULES IN QUBO SAMPLES

In this section, we adapt the traditional definition of association rule in our mining task. We describe the concept of equivalent alternatives, which is common in many problems. We also present a workflow and strategies to select promising rules from many candidates and apply them to simplify a QUBO problem.

## 3.1 DEFINITION

We first introduce the traditional definition of association rules in the field of Market Basket Analysis, and then adapt it into our QUBO samples mining task.

One famous application of ARM is to improve the business volume of a supermarket by making smart arrangement of items to racks, according to analysis on its customers' baskets (Market Basket Analysis, MBA). We use $\mathcal{X} = \{x_1, x_2, \ldots, x_m\}$ to represent all kinds of products or items in a supermarket. Let $\mathcal{C}$ be a set of transactions, where each transaction $c$ is a set of items such that $c \subseteq \mathcal{X}$. Suppose $P \subset \mathcal{X}, Q \subset \mathcal{X}$, and $P \cap Q = \emptyset$. A manager wants to understand if a customer is going to buy $Q$ if he/she has already put $P$ in his/her basket. An association rule $r$ is an implication of the form of $P \implies Q$. We say that the rule $r$ has support $s$ in the transaction set $\mathcal{C}$ if $s$ percent of the transactions in $\mathcal{C}$ contain $P \cup Q$. If a rule has a high support rate, it makes sense for the manager to place the racks of $P$ and $Q$ next to each other, to facilitate purchase behaviour.

We can find some correspondence between the traditional application of ARM and our use case in mining QUBO samples. $\mathcal{C}$ is the set of samples that we can collect upon solving a particular QUBO problem. Since each variable can take value $0$ or $1$, we interpret it as an indicator variable if a particular item is present. This corresponds to the idea that we also record the absence of an item in a transaction. Suppose that there are $m$ decision variables in a QUBO problem, then a sample to the QUBO problem is a bit string of length $m$. In addition, we define the length of a rule $|r| = |P| + |Q|$ to be the number of variables in the rule.

Generally, one can look for rules with high support. The variables suggested by a high-support rule are persistent in most of samples, and are likely to retain their values in the ground truth. Additionally, in an annealing system, we are also looking for a long rule, as a group of strongly associated variables is always more predictable than a single noisy variable. The definition of sample persistency Karimi & Rosenberg (2017) is a special case of our definition, where $|r| = 1$. Intuitively, we know that the high-support rules and the long rules are conflicting. The availability of rules that simultaneously meet the requirements of minimum length and minimum support is problem-dependent. We can excavate all representative rules in a set of samples to understand the availability of such promising rules. This raises the need for a mining rule with a wide range of support.

We use two examples to illustrate the concept of association rule mining for QUBO samples, which is shown in fig.1. We use a minimum support of $10\%, 20\%, \ldots, 80\%, 90\%$ in the mining task and organise all discovered rules in a plot, according to their support and length.

Fig.1a is a rule-rich example. The minimum length of the rules is about 60%, which means that about 60% of the variables are persistent in all samples. Some samples with extremely small error (in blue) share about 80% percent variables in common but are rare, since their support rate is under 20%. There are fewer circles in fig.1b, which suggest that the variables in the second example are not strongly associated. The cluster of circles in the lower right corner suggests that only 0.4-0.6% of the variables are strongly associated with each other with a support rate of 90-100%.

We also swap the solvers to let them work on a different problem and perform rule mining. Readers with interest can find the figures in appendix. The experiment indicates that QA tends to produce samples with a strong and diversified associations of variables, regardless of the topology of the problem. The classical solver DA has fewer and weak associations in variables, when the problem is large and sparsely connected.

## 3.2 EQUIVALENT ALTERNATIVES

The associations between variables could appear in the form of *Equivalent Alternatives*. We give a formal definition below.

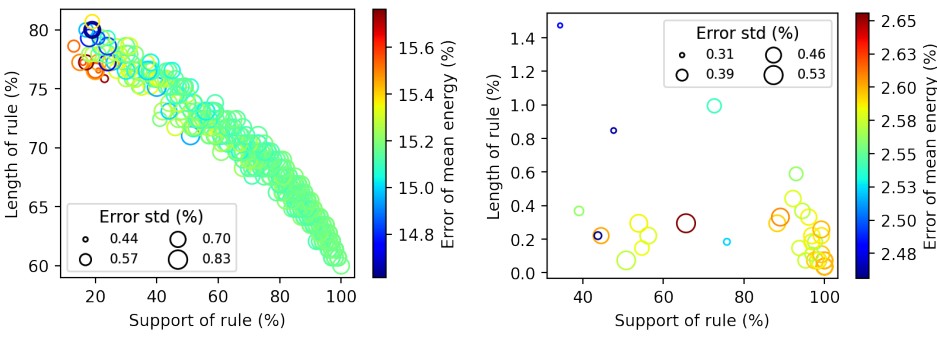

(a) D-Wave Pegasus Quantum Annealer  (b) Fujitsu 2nd Gen Digital Annealer

Figure 1: Rules discovered in the samples of different solvers. (a) D-Wave Pegasus Quantum Annealer on connectivity70, which is max-cut problem with 145 nodes and averaged degree of 70. X axis is the support of a rule in percentage. Y axis is the length of a rule, i.e., the number of variables in a rule. Each circle represents a subset of samples that follows the rule. The colour and size of a circle represents the mean and std of error, i.e., the difference in energy between the samples and ground truth. For example, a small blue circle represents some promising samples with small energy variance. (b) Fujitsu 2nd Gen Digital Annealer on pegasus2715, which is max-cut problem with 2715 nodes and sparse connectivity. The problem instances are described in details in section 5

Suppose $c_i$ and $c_j$ are samples for the same QUBO problem and are elements of $\{0,1\}^m$. They satisfy $f(c_i) = f(c_j)$, where $f(\cdot)$ is the cost function of the QUBO problem. Suppose that $\mathbb{A} : \{0,1\}^m \rightarrow \{0,1\}^m$ is a mapping function between samples. Given non-negative integers $p$ and $q$, if we can find a function $\mathbb{A}$ such that $\mathbb{A}^p(c_i) = c_j$ and $\mathbb{A}^q(c_j) = c_i$, and for all $c_i$, $f(\mathbb{A}^{p+q}(c_i)) = f(c_i)$, then we say $c_i$ and $c_j$ are equivalent alternatives. Here, $\mathbb{A}^p$ denotes the composition of $\mathbb{A}$ for $p$ times.

For the simplicity of explanation, let's first have a look at the famous max-cut problem, which is very popular among quantum computing research community. In max-cut problem, one wants to divide a set of nodes into two partitions, such that the sum of connection between the nodes from the two partitions are maximised. The conventional QUBO formulation for max-cut is shown in eq.(1).

$$\underset{x \in \{0,1\}^m}{\text{Maximise}} \sum_{i,j} w_{i,j} x_i (1 - x_j) \tag{1}$$

Each decision variable $x_i$ corresponds to a node. $w_{ij}$ is the connection or coupling between $x_i$ and $x_j$. $x_i = 0$ suggest the node $i$ is in the first partition, whereas $x_i = 1$ means the node $i$ is in the second partition. For max-cut, we can find a mapping function

$$\mathbb{A}(\{x_1, x_2, \ldots, x_m\}) = \{1 - x_1, 1 - x_2, \ldots, 1 - x_m\}$$

that converts any sample to its equivalent alternative. These alternatives have equal chance of being presented in sampling, since they have the same energy. The persistency of any variables is expected to be 50% in a fair sampling. Sample persistency Karimi & Rosenberg (2017) will loose its utility in this situation. One way to reveal the implicit persistency in max-cut samples is to find a long and high-support rule through ARM, and alternate samples such that they aligned with rule. We demonstrate the results of this method in Fig.1.

Equivalent alternatives are commonly seen in optimisation problems, such as number partitioning problem (two alternatives), max-cut problem (two alternatives), graph colouring ($T$ alternatives, where $T$ is the number of colours) and Traveling Salesman Problem (TSP) ($T$ alternatives, where $T$ is the number of cities). In TSP, two routes could be equivalent in order, but different in the starting cities. This is also referred to as Hamiltonian Cycles in the context of TSP and its Ising spin model Lucas (2014). A workaround to break the cycle in TSP is to add penalty in the energy function such that a solution is penalised if the travelling path does not start from a specific city. This workaround works fine for the classical annealing methods, but introduces additional penalty weight and severe analog control error to quantum annealing devices Pearson et al. (2019) and makes optimisation more challenging.

### 3.3 STRATEGIES AND WORKFLOW

Once we find the collection of rules, the next question is how do we select and apply the rules to simplify a QUBO problem.

We may ask which rule leads to a better improvement in the energy of the samples. Intuitively, we see that rules with high support are promising, as a variable with high persistency is likely to retain its value in an optimal state. We propose a rule selection strategy, which is called *Highest Support* (HS). HS sorts all rules according to their support. Optionally one can also sort the rules by their length as a second priority.

Intuitively, if a rule belongs to the best sample (e.g., with lowest energy for minimisation problem) in a set of samples, we might also see this rule to be very promising. We propose another rule selecting strategy, called *Lowest Energy* (LE). LE locates the neighbourhood of samples that follows a given rule, find the energy of the best sample within this neighbourhood, and use the energy as the key to sort rules. One can also use their support as a second key in the sort.

If we want to fix some decision variables in a QUBO, but the best rule suggested by HS or LE does not include enough number of variables, we can combine and apply the top $k$ rules in the list. But this could raise an issue when there are conflicts between these rules. For example a variable $x_i$ could appear as $x_i = 0$ in rule A and $x_i = 1$ in rule B. One can adopt a greedy strategy to keep the rule with top rank in the list, skip the conflicted rule with rank 2, and check if $r_3$ is free of conflict. In practice, the greedy strategy cannot always find enough number of variables to fix, due to conflicts. The selection of rules itself could become a maximum satisfiability problem. To simplify this issue, we just combine the top $k$ rules, with the conflicted variables taking the configuration in the top ranking rule.

To apply rules, fixing variables is the most straightforward way. By configuring a decision variable to 0 or 1, one can convert a linear term into a constant and convert a quadratic term into a linear term. This method is widely adopted in many other works Lewis & Glover (2016); Irie et al. (2021); Karimi & Rosenberg (2017). We follow the same way to apply the selected rules. A detailed explanantion on how to fix variables and reduce QUBO size can be found in appendix.

A typical workflow to facilitate optimisation can be as follows. We get a set of samples to a QUBO problem from a solver in the first run. To get more promising results, we find candidates of rules presented in the set of samples. We can select promising rules among candidates and fix variables accordingly to produce a reduced QUBO problem from the original. Then we pass the reduced problem to the solver for a second run and expect to find better results in the second run.

## 4 FAST ASSOCIATION RULE MINING FOR QUBO

As we mentioned in section 2.2, there is quite a few mis-match between the assumption of traditional ARM and that of our use case. The computational complexity of existing ARM algorithm would be intractable. We have to design our own algorithm to discover rules efficiently.

We propose an algorithm for mining QUBO samples, called Fast Association Rule Mining (FARM). as shown in Algorithm 1. FARM does not construct a data structure for each mining task. Instead, it directly finds "rules" in each sample and aggregate these rules to form the final output. The input of FARM is a set of samples. A sample is consisted of configurations to all decision variables in a QUBO problem. We use $c_i$ to represent the configurations of sample $i$. For each $c_i$, we find "Longest Common Rule" (LCR) in it by evaluating the Hamming distance, denoted by $h(\cdot, \cdot)$, between $c_i$ and its neighbourhood. We define LCR as the longest configurations that is shared by all the given samples and their equivalent alternatives. [1] The size of the neighbourhood is determined by the support $s$. A higher support results in a larger neighbourhood, and thus a shorter LCR shared by this neighbourhood. We aggregate the LCR of all samples and perform a de-duplication of the discovered rules at the end of the algorithm. The detailed implementation of finding LCR can be found in appendix.

---

[1]The definition of LCR should be distinguished from the famous Longest Common Subsequence (LCS) problem, as we do not assume order in decision variables. The occupancy of consecutive positions is not applicable to our definition of LCR.

**Input:** a set of samples $c_i, i \in [1, n]$ and support $s$
**Output:** rules
1 Set $\mathcal{K} = s \times n$;
2 Initialise rules as an empty list;
3 **foreach** $c_i$ **do**
4     Find $\mathcal{K}$ nearest neighbours of $c_i$ according to $h(c_i, c_j), i \neq j$;
5     Find longest rule $r_i$ in $c_i$ that shared by its neighbourhood ;
6     Calculate the actual support of the discovered rule;
7     Add $r_i$ to rules;
8 **end**
9 De-duplicate rules;
10 **return** rules

**Algorithm 1:** Fast Association Rule Mining for QUBO

Given a support rate, FARM returns rules with equal or greater support. This is achieved by only including a specific number of samples in the neighbourhood. To find rules with diversified support, one can call FARM with a range of support to produce a complete spectrum of rules just like the ones in fig.1. FARM is an approximated algorithm in the sense that it does not enumerate all rules, which is neither resource efficient nor necessary in our use case. We have demonstrated in fig.1 that FARM can find numerous amount of rules from the demo problem set. From them, we are only selecting one or two promising rules for subsequent evaluations. The approximation has a limited impact on the results.

## 5 EXPERIMENT

We evaluate our methods on two max-cut datasets with different topologies Huang et al. (2022). The connectivity-varied max-cut dataset is consisted of 32 graphs with 145 nodes in each graph. The averaged degree of graphs ranges from 1 to 140. The Pegasus-like max-cut dataset has ten graphs with a maximum of 12 edges attached to each node. The number of nodes ranges from 543 to 5430.

We include two solvers, D-Wave Pegasus Quantum Annealer and Fujitsu 2nd Gen Digital Annealer. As we demonstrated in section 3.1 quantum and classical solvers produce very different associations between variables. We include the two solvers to understand the generalisation of our methods. Regarding the hyper-parameter settings for the QA, we use the default geometric annealing schedule with 2000 $\mu s$ annealing time and 100 samples in each annealing task. We run DA in parallel tempering mode and set $1e6$ number of iterations with 128 replicas, which translates to 128 samples produced in annealing task.

We include sample persistency Karimi & Rosenberg (2017); Karimi et al. (2017) as a baseline. Heuristic methods like Chardaire et al. (1995) and Irie et al. (2021) are similar to sample persistency, in the sense that they do not take associations between variables into account. We do not include them in the comparison.

We implement FARM algorithm in Python. It is able to find rules with a few hundreds of variables within seconds. Readers with interests can find the plot of timing and explanation in the appendix.

### 5.1 QUANTUM ANNEALER ON CONNECTIVITY-VARIED PROBLEMS

We first evaluate our methods with quantum annealer on connectivity-varied problems. Quantum Annealer (QA) is generally weak on densely connected graphs. Existing work suggests prominent performance degradation in QA when the average degree is greater than 12. Fixing variables can effectively reduce averaged degrees. In fig.2, we demonstrate the result on the connectivity70 max-cut problem, in which averaged degree is 70.

In fig.2a, the blue bars represent the histogram of variable persistency measured by the method in Karimi & Rosenberg (2017), which suggest the values of the samples are quite random. For example, it says that more than 80% of the variables have a persistence of 50%. However, we have already demonstrated the association rules discovered by FARM in fig.1a, which suggests that there is quite a

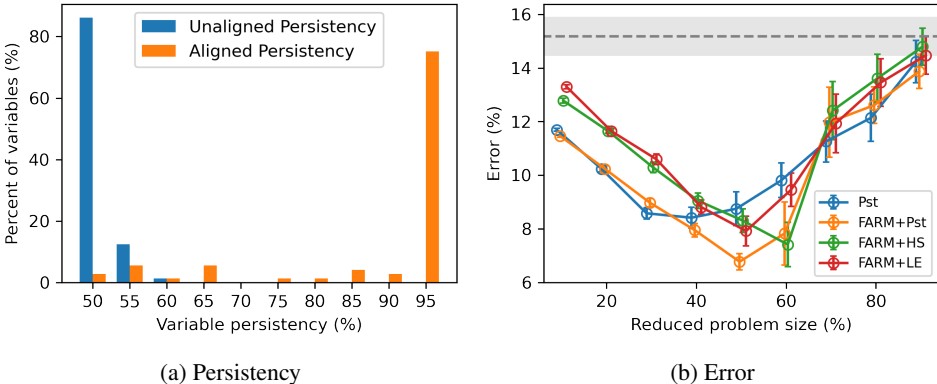

(a) Persistency

(b) Error

Figure 2: Quantum Annealer on the connectivity70 max-cut problem. (a) Variable persistency in the results of first run, with and without alignment using the rules discovered by FARM (b) Error of different strategies. X axis represents the size of QUBO in the second run, in percentage. Y axis represents error between achieved energy and ground truth, in percentage. Lower error is better. The dotted line and gray area represent the mean and std of the error in first run. The curves with error bars represent the statistics of errors in second run.

level of persistency in the samples. We can recover the persistency by replacing some samples with their equivalent alternatives. From the candidates for the rules discovered by FARM, we select the longest rule with 100% support and alternate the samples so that they align with the rule. With the alignment, about 75% of variables have persistent value across all samples.

Fig.2b demonstrates the performance of different strategies. Pst is the strategy used by Karimi & Rosenberg (2017), which sorts the variables by persistency and fixes the top $k$ variables. FARM+Pst first align the samples and then apply Pst accordingly. FARM+HS and FARM+LE combine FARM with the HS and LE strategy, which we proposed in Section 3.3. Fig.2b suggests that FARM+HS achieves a lower error compared to Pst and FARM+LE, but the improvement is limited. We can see that the FARM+HS curve in green at 40% has a higher error than that of Pst in blue. Despite fig.1a suggesting that we can find a rule which has 100% support and includes 60% of variables of the original problem, applying such a promising rule does not give FARM+HS advantage over other strategies, which is not as we expected. This could be due to the manufacturing limitation and imperfection in the analog control in the quantum device Pearson et al. (2019), which causes a mis-match between the intended problem and the actual implementation of the problem. In other words, the promising rule that we discovered corresponds to the mis-implemented problem, not the original one. In comparison, FARM+Pst has the best performance, as it achieves competitive error as low as that of FARM+HS, and retain a low error in a wide range of reduced problem size.

## 5.2 DIGITAL ANNEALER ON PEGASUS-LIKE PROBLEMS

We evaluate our methods with Fujitsu 2nd Gen Digital Annealer (DA) on Pegasus-like max-cut problems. DA is an efficient hardware implementation of the parallel tempering algorithm and is specifically optimised for densely connected problems. DA is weak on sparsely connected problems, like the ones in Pegasus-like max-cut problems. Reducing variables raises the sparsity of the problem and potentially pose challenges to DA. In fig.3, we demonstrate the result on the pegasus2715 max-cut problem, in which there are 2715 nodes.

Fig.3a suggests that there is very limited persistency in the samples even if we apply alignment to the samples. This match with our observation in fig.1b, where the longest rule found by FARM only includes 1.4% variables of the QUBO problem.

With such a low level of persistency, Pst and FARM+Pst cannot find promising results. Their curves at 90% problem size lead to 6% averaged error in the second run, which is about 3% worse off compared to the results in the first run. As the number of fixed variables increases, their curves go off the chart and reach about 35% averaged error when the problem size is reduced to 10% of the original

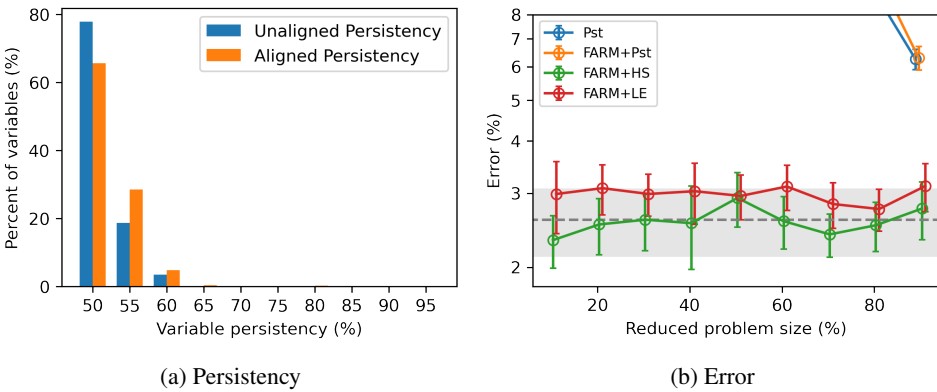

(a) Persistency

(b) Error

Figure 3: DA on the pegasus2715 max-cut problem. The plot setting is the same as fig.2

size. In comparison, FARM+HS and FARM+LE achieve similar error in the second run. FARM+HS is slightly better than FARM+LE and can sometimes outperform the results in the first run.

Given the same problem instance pegasus2715, if we compare the rules discovered in the sample set produced by QA, we know that the length of rules ranges from 10-60%, which is much longer than those for DA and indicates the potential space of improvement for QA. We do not repeat the experiment of QA on pegasus2715, since QA is already able to find global optimal in a single run Huang et al. (2022). The comparison between QA and DA on pegasus2715 suggests a stronger potential of association rule for QA.

## 6 CONCLUSION AND DISCUSSION

In this paper, we define the association rule mining in the context of the mining of QUBO samples. We propose strategies to select promising rules and apply them to simplify QUBO problems. We propose Fast Association Rule Mining (FARM), which effectively finds rules of a wide range of support from QUBO samples. Experiments suggest that our method is aware of associations among variables and can simplify QUBO problems and improve the optimisation results.

Determining the number of variables that maximizes performance is challenging and remains an open question in the literature. We demonstrate FARM as a visualisation tool that help user understand the distribution of the rules, and narrow down the range of optimal number of variables to be fixed. In the history of this research topic, we are the first to investigate the distribution of frequent patterns in terms of length v.s. support rate. Our next step is to examine the corresponding determination methods.

We say the rules discovered by FARM are correct in the sense that they do exist in the sample set. But the discovered rules do not necessarily hold in the global optimal, even if they have 100% support rate in the sample set. This is due to the noises and errors Pearson et al. (2019) in true quantum annealing devices. We do not investigate the divergence between the distribution of sample set and the solution space of the original problem, as this is out of the scope of this work. In our future work, we are going to investigate how the divergence impact the association rule mining for QUBO.

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

## A  LIST OF SYMBOLS

Table 1: List of symbols

| | |
|---|---|
| $x$ | A decision variable |
| $v$ | The binary value of a decision variable |
| $m$ | Number of samples |
| $n$ | Number of decision variables |
| $T$ | Number of equivalent alternatives to a sample |
| $w_i, w_{i,j}$ | Weight of linear terms and quadratic terms |
| $c$ | A sample or configuration |
| $\mathbb{c}$ | The equivalent alternatives of a configuration |
| $r$ | An association rule |
| $\mathbb{r}$ | The equivalent alternatives of an association rule |
| $h(\cdot, \cdot)$ | hamming distance between two rules or two configurations |
| $s$ | support |

## B  FIND LONGEST COMMON RULE (LCR)

To find the Longest Common Rule (LCR) in a sample $c_i$, we use another customised algorithm, as shown in Algorithm 2. For each $c_i$, we use the XNOR operator and compare it with the rest of the samples and their equivalent alternatives, denoted as $\mathbb{cfg}_j$. $A_{i,j,k}$ is a bit array, whose size equals to the number of decision variables in the QUBO problem. A one in $A_{i,j,k}$ suggests that the corresponding decision variable in the two samples shares the same value. The Hamming distance between $c_i$ and $c_j^k$ can be calculated from $A_{i,j,k}$. Among all the alternatives of a sample, we only keep the one closest to $c_i$. As we dump the rest alternatives, we can rename $A_{i,j,k}$ as $A_{i,j}$. $A_{i,j}$ for all $j$ is a bit map that marks all common values shared by $c_i$ and its neighbourhood. We can find variables that are persistent throughout the neighbourhood, by applying aggregation based element-wise AND

**Input:** a reference configuration $c_i$ and a set of $c_j, j \in \mathcal{K}$
**Output:** rule $r$
1 **foreach** $\mathbb{c}_j, i \neq j$ **do**
2     **foreach** $c_j^k \in \mathbb{c}_j$ **do**
3        $A_{i,j,k} = \text{XNOR}(c_i, c_j^k)$ ;
4        $h(c_i, c_j^k) = \text{sum}(\mathbb{1} - A_{i,j,k})$;
5     **end**
6     Keep $c_j^k$ with shortest $h$, and name it $c_j$;
7     Dump the rest alternatives and rename $A_{i,j,k}$ as $A_{i,j}$;
8 **end**
9 Aggregate $A_{i,j}$ by applying element-wise AND along subscript $j$;
10 Select decision variables whose corresponding bit in the aggregation is 1;
11 Selected variables and their values in $c_i$ form a rule $r$;
12 **return** $r$

**Algorithm 2:** Find the LCS in a sample $c_i$

operation along subscript $j$. Those persistent variables and their values forms the longest common rule within this neighbourhood.

The complexity of the FARM algorithm is $O(m^2 n T)$, where $m$ is the total number of samples in a mining task, $n$ is the total number of variables in a sample, $T$ is the total number of equivalent alternatives to each sample. The majority of the calculations in FARM are logical operations like XNOR or AND. There are limited levels of data dependency between these calculations, which means that we can exploit massive parallelism, e.g., like vectorisation or SIMD in advanced computing architecture. The complexity of FARM scales linearly with $n$, suggesting its ability to handle a mining task with a large number of decision variables. Since FARM is expecting a limited number of samples in a mining task, the quadratic complexity factor $m^2$ is still tractable for modern computing architecture.

## C APPLY A RULE TO REDUCE QUBO PROBLEM

A QUBO minimisation problem can be written in the form of $\min_{x \in \{0,1\}^n} x^T Q x$, where $x$ is a vector of decision variables, $Q$ is the QUBO matrix. We can also rewrite the QUBO in the following representation.

$$\min_{x \in \{0,1\}^n} \sum_i w_i x_i + \sum_{i,j} w_{ij} x_i x_j + C \tag{2}$$

$w_i$ is on the diagonal of $Q$, $w_{i,j}$ is the off-diagonal element of $Q$. We refer to $w_i x_i$ as a linear term, and $w_{ij} x_i x_j$ as a quadratic term. $C$ is the constant. If we discover through FARM that a variable $x_j$ equals to one in most of samples, we can assign 1 to $x_j$. This will change the linear term $w_j x_j$ into a constant $w_j$, and change the quadratic term $w_{ij} x_i x_j$ into a linear term $w_{ij} x_i$. If $x_j$ equals to zero in most of samples, we can assign 0 to $x_j$, such that the linear term $w_j x_j$ and $w_{ij} x_i x_j$ become zero. In either case, the number of variables in the QUBO problem reduces.

In our experiment, we use the function "fix_variables" provided by D-Wave Ocean 5.5, which is the solver API used to access the quantum annealer.

## D EXPERIMENT APPENDIX

### D.1 THE DISCOVERY OF RULES

### D.2 TIMING OF FARM

The timing of FARM is shown in fig.5.

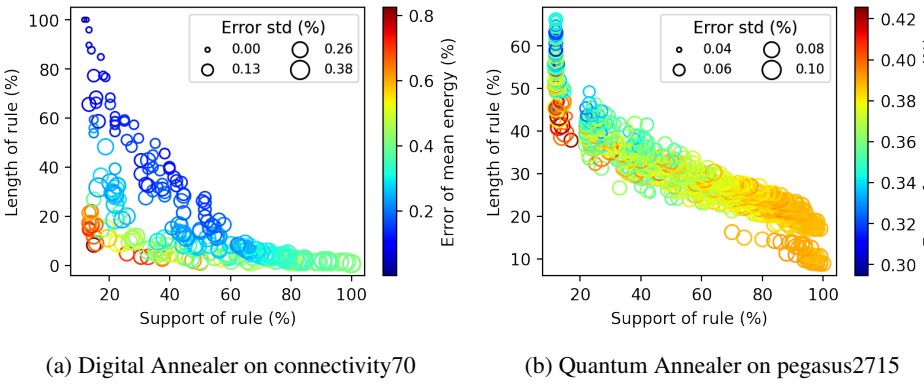

(a) Digital Annealer on connectivity70

(b) Quantum Annealer on pegasus2715

Figure 4: Frequent patterns found by different solvers

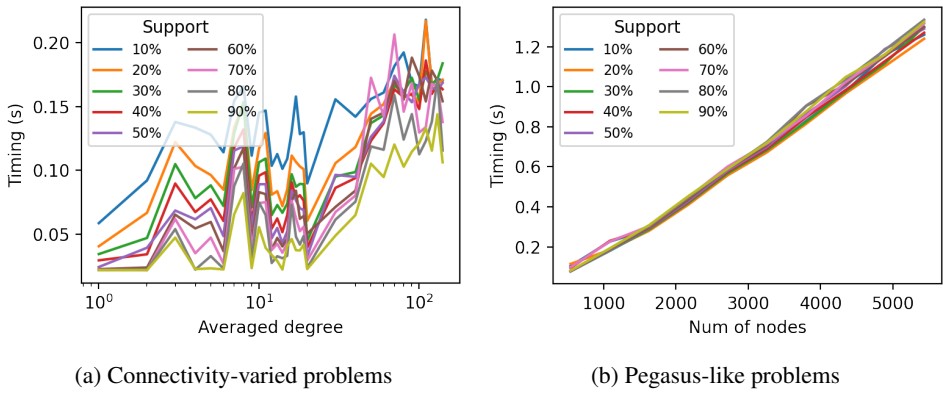

(a) Connectivity-varied problems

(b) Pegasus-like problems

Figure 5: Timing of FARM with respect to problem complexity and support

Fig.5a shows the timing of FARM on the sample set produced by QA on the connectivity70 problem. The run-time increases with the number of averaged degrees. It also increases as we use a lower support in the mining. Fig.5b shows the timing of FARM on the sample set produced by DA on the pegasus2715 problem. The run-time linearly increase along with the number of nodes. The run-time is not sensitive to the setting of support, which match our expectation because we find limited number of rules as shown in fig.1b.

