# OpenReview forum: "Association Rules in QUBO Samples and Where to Find Them"
_ICLR.cc/2023/Conference — Submitted to ICLR 2023_

### Official Review · Reviewer_ABPP · 2022-10-24

**Confidence:** 4
**Correctness:** 3
**Technical Novelty And Significance:** 2
**Empirical Novelty And Significance:** 2
**Recommendation:** 3

**Clarity, Quality, Novelty And Reproducibility:**

Some concepts introduced in the paper should be clarified. It seem an incremental work where many concepts could better described and the results should be improved adopting different strategies.

**Strength And Weaknesses:**

+ An interesting idea compared to the state of the art works
- Poor results

**Summary Of The Paper:**

The paper proposes to use heuristic method to discover association rules between variables in an optimization problem.


**Summary Of The Review:**

The introduction of the problem is unclear. For instance in the introduction in not clear the the meaning of samples in an optimization problem. This should be clarified.

The are same part of the paper that could be better explained. For instance how the rules have been adopted to reduce the size of the dataset is not reported.

The idea could be interesting but the experimental evaluation is poor and the validity of the proposed approach is non confirmed by the reported results.

---

> ### Author Response · Authors · 2022-11-18
> **Revise the paper and add clarifications according to reviewers comments**
>
> C1
> Some concepts introduced in the paper should be clarified. It seem an incremental work where many concepts could better described and the results should be improved adopting different strategies.
>
> R1
> This work investigates the association between decision variables in annealing-based solvers. We do not see similar work in the literature. We do not see our work as an incremental one.
> Sampling is a core concept in a probabilistic optimisation method, where you sample a lot of times to find a promising solution. For those readers who are not familiar with annealing-based method, we add a brief introduction in the first section for readers to quickly pick up necessary information.
> Annealing, e.g., simulated annealing, is a family of probabilistic methods for optimising the variables of a system (e.g. minimising a function). In annealing, we heat the system to high temperature level and gradually bring down the temperature. Variables in the system gradually lose their energy and eventually sit in a low energy state. One round of heating and cooling, i.e., an annealing process, produces one sample, which is a configuration of all variables. Annealing process is a random process. The energy distribution of the samples follows a Boltzmann distribution, i.e., the system is more likely to be in a lower energy state. Quantum annealing works in a similar way, except that it makes use of quantum mechanics.
>
> C2
> The are same part of the paper that could be better explained. For instance how the rules have been adopted to reduce the size of the dataset is not reported.
>
> R2
> As we briefly mentioned in section 2 and 3.3, we use rule to condition a QUBO problem, such that the number of variables is reduced. We have never mentioned about reducing the size of a dataset. The way of adopting rules is well documented in previous literature. To help readers better understand how it works, we add detailed explanation in the appendix.
>
> C3 and C4
> poor result
> The experimental evaluation is poor and the validity of the proposed approach is non confirmed by the reported results.
>
> R3 and R4
> Section 5.1 demonstrates our method outperforms SOTA on Quantum annealing (QA). The experiment on Digital Annealer (DA) in section 5.2 demonstrates the limited utility of "fix variables" for classical solvers. Section 5.1 and 5.2 indicate the critical difference between QA and DA on this topic, which missing in existing literature.
> We add clarification accordingly in section 1 and 5.

---

### Official Review · Reviewer_5aqV · 2022-10-24

**Confidence:** 2
**Correctness:** 4
**Technical Novelty And Significance:** 3
**Empirical Novelty And Significance:** Not applicable
**Recommendation:** 6

**Clarity, Quality, Novelty And Reproducibility:**

Clarity: OK, but sometimes a bit difficult to follow for those who are not familiar with a QA.

Quality: Fine, though the proposed method is a simple heuristics.

Novelty: Combination of annealer and rule mining would be novel

Reproducibility: OK.


**Strength And Weaknesses:**

Strength:
- The basic idea, combining a quantum annealer (QA) and association rule mining, is interesting and seems novel.
- The empirical results suggest superiority of the proposed method.

Weaknesses:
- As the authors mentioned, determining the number of variables that maximizes performance is difficult.
- No theoretical guarantee or justification for the approximation quality (e.g., risk to find wrong rule) is provided.
- The paper is sometimes difficult to follow for those who are not familiar with a QA.

**Summary Of The Paper:**

The paper proposes applying association rule mining for reducing the problem size of QUBO. The mining is performed for solution samples, and the authors proposes an efficient approximate association rule mining specifically for QUBO. Empirical results show that the proposed procedure can substantially reduce the problem size and find good solutions compared with an existing method.

**Summary Of The Review:**

The paper seems to try an interesting approach combining QA and rule mining. Although the proposed method is a simple heuristics that do not have a theoretical support, the empirical results suggest the practical usefulness of the proposed framework. However, to be honest, I am not familiar with the topic and my review contains educated guess.

---

> ### Author Response · Authors · 2022-11-18
> **Revise the paper and add clarifications according to reviewers comments**
>
> C1
> As the authors mentioned, determining the number of variables that maximizes performance is difficult.
>
> R1
> Determining the number of variables is challenging and remains an open question and a weakness in all existing literature. We demonstrate FARM as a visualisation tool that help user understand the distribution of the rules and narrow down the range of optimal number of variables to be fixed. In the history of this research topic, we are the first to investigate the distribution of frequent patterns in terms of rule length v.s. support rate. Our next step is to examine the corresponding determination methods.
>
> We add this clarification to the conclusion and discussion section
>
> C2
> No theoretical guarantee or justification for the approximation quality (e.g., risk to find wrong rule) is provided.
>
> R2
> FARM is an approximated algorithm in the sense that it does not enumerate all possible rules, which is neither resource efficient nor necessary in our use case. We have demonstrated that FARM can find numerous amounts of rules from the demo problem set. From them, we are only selecting one or two promising rules for subsequent evaluations. The approximation has a limited impact on the results.
> We add this clarification in the description of FARM algorithm.
>
> C3
> The paper is sometimes difficult to follow for those who are not familiar with a QA.
>
> R3
> We add a brief intro of annealing in the first section to help readers quickly pick up the information necessary for understand the background.
>
> Annealing, e.g., simulated annealing, is a family of probabilistic methods for optimising the variables of a system (e.g. minimising a function). In annealing, we heat the system to high temperature level and gradually bring down the temperature. Variables in the system gradually lose their energy and eventually sit in a low energy state. One round of heating and cooling, i.e., an annealing process, produces one sample, which is a configuration of all variables. Annealing process is a random process. The energy distribution of the samples follows a Boltzmann distribution, i.e., the system is more likely to be in a lower energy state. Quantum annealing works in a similar way, except that it makes use of quantum mechanics.

---

### Decision · Program_Chairs · 2023-01-20

**Decision:**

Reject

**Justification For Why Not Higher Score:**

Results are preliminary.

**Justification For Why Not Lower Score:**

The idea is worth exploring.

**Metareview: Summary, Strengths And Weaknesses:**

We apologize for failing to find a third reviewer despite many requests and reminders.

As the area chair, I am familiar with quantum annealing and with mining association rules. I agree with the overall sentiment of the two reviewers that this paper is interesting. It should be published, but not at ICLR now.

In their responses, the authors provided clarifications but did not show that the reviews are incorrect. To help the authors resubmit elsewhere, here are a few questions or suggestions.

1. Clarify whether you used actual quantum hardware from D-Wave and/or Fujitsu. If yes, show that your method enables finding optimal solutions better or faster using the hardware. None of the current figures show this.

2. The FARM method does not have guarantees, and the rules discovered may be false in the solution to the annealing problem. So the contribution of the paper is not solid and strong enough for ICLR.

3. Be more incisive when discussing previous research. This area chair has been a professor and FAANG director in ML for over 30 years. I am not aware of a single real-world useful application of association rules, for market basket analysis or for anything else.

Also, fix the formatting of citations, which has words run together with missing spaces